# Soil Moisture Conservation through Crop Diversification and Related Ecosystem Services in a Blown-Sand Area with High Drought Hazard

**DOI:** 10.3390/plants13040494

**Published:** 2024-02-09

**Authors:** Dénes Lóczy, József Dezső, Tamás Weidinger, László Horváth, Ervin Pirkhoffer, Szabolcs Czigány

**Affiliations:** 1Department of Physical and Environmental Geography, Institute of Geography and Earth Sciences, University of Pécs, Ifjúság Útja 6, 7624 Pécs, Hungary; dejozsi@gamma.ttk.pte.hu (J.D.); pirkhoff@gamma.ttk.pte.hu (E.P.); sczigany@gamma.ttk.pte.hu (S.C.); 2Department of Meteorology, Institute of Geography and Earth Sciences, Eötvös Loránd University, Pázmány Péter Sétány 1/a, 1117 Budapest, Hungary; weidi@caesar.elte.hu; 3HUN-REN-SZTE Photoacoustic Research Group, Department of Optics and Quantum Electronics, University of Szeged, Dóm Tér 9, 6720 Szeged, Hungary; horvath.laszlo.dr@gmail.com

**Keywords:** crop diversification, wind erosion, organic matter, water retention, carbon storage, greenhouse gas emissions, asparagus, cover crops

## Abstract

Soil moisture reserves are a key factor in maintaining soil fertility and all other related ecosystem services (including carbon sequestration, soil biodiversity, and soil erosion control). In semiarid blown-sand areas under aridification, water preservation is a particularly crucial task for agriculture. The international Diverfarming project (2017–2022), within the EU Horizon 2020 Program, focused on the impacts of crop diversification and low-input practices in all pedoclimatic regions of Europe. In this three-year experiment conducted in the Pannonian region, the impact of intercropping asparagus with different herbs on some provisioning and regulating ecosystem services was evaluated in the Kiskunság sand regions. Relying on findings based on a range of measured physical and chemical soil parameters and on crop yields and qualitative properties, advice was formulated for farmers. The message drawn from the experiment is somewhat ambiguous. The local farmers agree that crop diversification improves soil quality, but deny that it would directly influence farm competitiveness, which primarily depends on cultivation costs (such as fertilization, plant protection, and labour). Further analyses are needed to prove the long-term benefits of diversification through enriching soil microbial life and through the possible reduction of fertilizer use, while water demand is kept at a low level and the same crop-quality is ensured.

## 1. Introduction

In the temperate belt, monoculture in crop cultivation has led to the widespread degradation of fertile land and landscape diversity [1], and has resulted in enhanced economic risks [2] concerning the food supply. At the same time, heavily modified agricultural landscapes have also become vulnerable to climate change. Blown-sand mantled alluvial fans, with a particularly unfavourable water balance, are as vulnerable in this respect as any wetland [3].

Presently, desertification in the tropical belt and aridification at mid-latitudes [4,5] are viewed as major climate change-related hazards to the well-being of humankind. Deserts and desertification-affected areas extended over 41.3% of the Earth’s land area in 2000 [6]. The largest desert, the Sahara—for instance—is expanding at a rate of 11,000 km^2^ y^−1^ [7], and the arid areas of other continents show similar trends. The updated FAO estimate for the world desertification rate is 120,000 km^2^ y^−1^ [5]. Extratropical areas, such as Hungary, are also affected [8].

The Farm to Fork strategy of the European Union, aiming for a fair, healthy, and environmentally-friendly food system [9], is to also be implemented in arid lands. It includes concepts on which measures directed at making crop cultivation more sustainable and at increasing diversity can be based [10]. Crop diversification may be an important pillar of this strategy. As defined, crop diversification means that plants at different developmental stages are present in the same field at different times with the purpose of improving the level of various ecosystem services. Crop rotation, strip harvesting, genetic diversity, and weed strips in monoculture, polyculture, agroforestry, and mixed use of the landscape have long been applied and are efficient means of diversification [11]. Intercropping can be added to the list as a further solution.

Some potential, but easily realizable, benefits of farm-level crop diversification in multiannual crop plantations are [12]:It increases incomes for small farms;Helps withstand fluctuations in commodity prices;Provides resilience to extreme weather conditions (surplus rainfall or drought);Reduces the costs of production;Provides more varied and healthy foods, both for humans and livestock;Diminishes pest hazards and weed growth;Enhances beneficial pollinator populations;Improves soil quality in the long term;Broadens employment.

Diversification, however, can also be interpreted in a broader sense. Some ecologists are convinced that both wild and cultivated plants are sociable beings and prefer each other’s company to living alone (i.e., in monocropping) [13,14]. Through the optimal selection of cover crop diversification, there may be socio-economic benefits as well—presenting itself as the best way of exploiting natural potential [15,16].

The Diverfarming project (2017–2022), within the EU Horizon 2020 program, aimed at the evaluation of the impact of crop diversification and low-input interventions [17]. Diverfarming is intended to enhance the long-term resilience, sustainability, and economic revenues of agriculture across the EU by assessing the real benefits and minimising the limitations, barriers, and drawbacks of diversified cropping systems using low-input agricultural practices that are tailor-made to fit the unique characteristics of six EU pedoclimatic regions (the southern and northern Mediterranean, Atlantic, continental, Pannonian, and boreal regions). The case studies also covered a wide range of cultures from perennial woody plants such as citrus crops, olive plantations, and vineyards as well as annual fruits such as sugar melons, vegetables (tomato, peas), cereals, fodder crops, etc. Other Horizon 2020 projects have also been directed at the better provision of ecosystem services (soil fertility, carbon sequestration, reduction of greenhouse gas emissions, soil erosion and contamination) in Europe [18,19].

In the most comprehensive overview of the topic, Beillouin, D. et al. [16] analysed the available literature (95 meta-analyses from 85 countries) to assess the effects of crop diversification in agroecosystems. They found that crop diversification significantly enhanced crop productivity (median effect + 14%) and associated biodiversity (+24%, i.e., the biodiversity of non-cultivated plants and animals). In addition, several supporting and regulating ecosystem services including water quality (+51%), pest and disease control (+63%), and soil quality (+11%) were also improved.

Members of the Diverfarming consortium collected information on stakeholders’ perceptions of barriers and opportunities for implementing farming practices and crop diversification strategies in intensive rainfed and irrigated cereal-based cropping systems in Italy [20]. To make the introduction of diversification solutions easier, they selected cover crops which are already cultivated there as monocultures and are adapted to the local pedoclimatic conditions.

In Hungary, the area hardest hit by drought hazards is the Kiskunság sandy region in the centre of the Danube–Tisza Interfluve [4,21,22]. The situation is exacerbated by the constantly dropping groundwater levels [8]. Sporadic years with higher-than-average precipitation could disrupt but not turn the aridification trend [23]. Reduced soil moisture contents on the surface provide favourable conditions for wind erosion, which further reduces soil fertility, which is already low anyway [24,25]. Consequently, in the Kiskunság, the environmental problems of land use equally reflect unfavourable climatic and soil properties (Figure 1).

In the study area of the Diverfarming experiment, a multiannual horticultural crop—asparagus *(Asparagus officinalis* L.)—is grown. The broad alleys between asparagus rows allow for the application of intercropping as a means of crop diversification. The intercropped cover crops were assumed to promote soil moisture retention, enrich the soil in organic matter with residues (green manuring), and reduce wind velocity (and thus, soil erosion) through enhanced surface roughness.

In addition, low-impact interventions are also used in asparagus cultivation involving the application of ’greensoil’ (granulate organic matter mixed with mineral fertilizer) and (from November to March) cellulose-decomposing bacteria, integrated pest control, and regulated irrigation. In the study area, regulated deficit irrigation was applied—an optimization strategy in which irrigation water is only added to the crop during its drought-sensitive growth stages and in proportion to the missing rainfall amount.

Several ways of sustainably combating drought and wind erosion hazards have been proposed [26]:Enhancing soil moisture preservation by preventing or hindering desiccation. This would require regular irrigation to allow crusting, but under the climatic conditions of the study area, it cannot be implemented in the long run.Ensuring better protection of the surface by vegetation cover, reducing wind energy by forest belts [27], or providing a contiguous cover by denser cropping. However, this is jeopardized by the water scarcity almost permanently present throughout the growing season.Mulching could increase the soil’s organic matter content, and thus, soil aggregation. Organic matter in itself contributes to soil moisture conservation and balances soil temperature and pH. However, for the decomposition of organic matter via microbial activity [28], the soil moisture content available in the long term is usually not adequate. The aggregate stability of the wind-blown sand is also low because of the minimum clay content (the latter is only higher in buried paleosols, formed in wetter spells of landscape history).

In the international literature, numerous techniques have been proposed to combat wind erosion and, at the same time, increase soil fertility (organic matter content). In a pilot study in Kuwait, Burezq, H. [29] stabilized native blown-sand soil (T1) with biochar and animal manure (T2) and T3 with biochar, animal manure, Urea Formaldehyde (UF), Sulfonated Naphthalene Formaldehyde (SNF), and Polyvinyl Alcohol (PVA) (T3). The erosion rate of native sandy soil (T1) has increased from 3.33 to 4.77 to 7.35 g m^−2^ min^−1^ when wind speed was increased from 5 to 10 and 15 m s^−1^, respectively. At the same wind speeds, the measured erosion loss was 1.99, 3.07, and 5.32 g m^−2^ min^−1^ for T2 and 1.17, 2.6, and 4.24 g m^−2^ min^−1^ for T3. In asparagus fields with more cohesive soil in the UK, Niziolomski, J. C. [30] found compost application to be less effective than straw. With straw mulch applied at 5 t ha^−1^, overall soil loss was reduced by 72%, with soil erosion rates just above 1.4 t ha^−1^ y^−1^—a tolerable value in the EU.

The main hypothesis of the whole Diverfarming project was that crop diversification has environmental benefits, as it contributes to the improvement of a whole range of ecosystem services (as listed in [12]). More specifically, the objective of the present paper was to point out opportunities for higher soil fertility, better soil moisture conservation, and wind erosion prevention through crop diversification in a severely drought-stricken region. The socioeconomic implications of crop diversification are not investigated here.

## 2. Study Area

A unique feature of the Kiskunság landscape used to be its high landscape-level diversity and the mosaical pattern of extremely dry sand dunes, alternating with saline wetlands between them. The contrast between these habitat types, however, is getting more and more subdued [31], and the landscape pattern is becoming simpler.

In the physico-geographical divisions of Hungary [32], the study area in the outskirts of the village Jakabszállás belongs to the alluvial fan of the Danube, right on the southern boundary between the Kiskunság and Bugac sand regions (Figure 2). It has calcareous blown-sand soil (in the WRB classification system: Calcic Arenosol); its environmental properties are extremely unfavourable for cultivation [33,34]—the sand fraction is predominant at 90–99%, while the clay content is minimal (0–0.3%). A low organic matter content (total organic carbon: 1.5–3.5 g kg^−1^) and the virtual absence of particles in the >250 μm diameter fraction hinders aggregation. Inorganic carbonates are found in medium amounts (3.5–4.5%). Soil fertility and the choice of agricultural crops are first of all limited by the low soil moisture content, as well as its extreme seasonal distribution and the low water retention capacity of the topsoil [35,36,37]. Frequent desiccation exposes the soil to sandstorms, which carry away large amounts of low-density humus particles.

## 3. Results and Discussion

The efficiency of diversification heavily depended on the soil coverage provided by the different options. Utilizing water from sporadic showers, pea and oat ensured an almost perfect (>85%) cover from the second year of the experiment on. Unfortunately, this positive effect of the cover crops was limited to a relatively short period, from late April to early June. This means that in early spring, when strong winds usually blow, the ground remains uncovered.

Soil cover and wind erosion risk are affected by the frequency of total soil desiccation (soil moisture content: <0.02 cm^3^ cm^−3^) as well as precipitation (Figure 3). Fortunately, for a longer spell in the second year of the experiment, the soil moisture reached a satisfactory value (0.1 cm^3^ cm^−3^; Figure 4). For the conservation of soil moisture, diversification D1 (asparagus + pea) was pre-eminent, especially in the deeper soil layer (Figure 4).

The achieved nutrient contents showed a variable picture. The total N content in the uppermost 10 cm soil layer amounted to 310 mg kg^−1^ on average, with peaks in the first year when more fertilizers were applied. Over the rest of the experimentation period, no significant differences were observed between the diversifications D1 and D2 in this respect. In the deeper 10–30 cm layers, the varying intensity of microbial activity generated a great variation in nitrogen supply; it was 60 mg kg^−1^ in asparagus monocropping (M), 110 mg kg^−1^ in D1 (pea), and as low as 50 mg kg^−1^ in D2 (oat).

In the diversifications, pH values showed a gradual decrease. With similar rates of fertilization, this could be explained by the growing acid release by the roots of the developing cover crops. Electrical conductivity remained at high levels due to the chemical treatments of crops. The cation exchange capacity (CEC) and the sum of bases values rose over the three-year experiment and indicated some improvement in the nutrient supply (Figure 5a). The available phosphorus reserves peaked in 2020 and in the diversified plots (particularly in the 0–10 cm soil layer—Figure 5b).

As had been expected, the competition between the main crop (principal root zone at 20–120 cm depth) and the cover crops (at 0–20 cm for pea and 0–70 cm for oat) for moisture and nutrients was minimal. Although very low clay contents (low aggregation potential) exclude the enrichment of the soil with nitrogen, the total organic carbon content could be raised by adding crop residues to the soil (however, the decomposition rate of such residues is low in dry environments).

What was the regularity in the fluxes of greenhouse gases? In winter emissions, which depend on temperature, fluxes were negligible (Figure 6). Peaks of N_2_O coincided with fertilization events (nitrification of ammonia) and higher CO_2_ emissions. Total CO_2_ emissions amounted to 20.7 t ha^−1^ and N_2_O emissions to 14.5 kg ha^−1^ over the three years. Since the global warming potential (GWP) of N_2_O 300-fold exceeds that of CO_2_, its total CO_2_-equivalent emission is 4.36 t ha^−1^; therefore, the total greenhouse gas emission of the soils was 25.0 t ha^−1^ in CO_2_ equivalent. These findings are in harmony with other low CO_2_ emission values above low vegetation [38].

Soil is a sink for methane. The CH_4_ absorption rate is low compared to CO_2_ emissions—only −3.68 kg ha^−1^ over the three years, which is merely 0.05% of CO_2_ emissions (in carbon content).

Nitrogen-fixing bacteria and mineral fertilizer application led to increased N_2_O emission in D1 plots (with pea). These findings, however, are not in accordance with data from the literature [39,40], where an opposite trend was held probable for legumes. The higher nitrogen content of the soil of the D1 plots cannot explain fully the enhanced N_2_O emission, because D2 plots also had higher emission levels due to the incorporation of cover crop residues into the soil [41]. Our results are supported by Basche, A. D., et al. [42], who claim that leguminous cover crops may increase, while other herbs may decrease, N_2_O emissions. The literature also indicates that where organic cultivation is possible, lower greenhouse gas emissions [43], a longer shelf life, and the higher quality of asparagus spears [44] can be achieved.

From the erosion measurements, the vertical profile of the aeolian transport was defined. The amount of sand saltated 15 cm above the ground surface reached an average of 2130 kg y^−1^ over the experimentation period. At a 200 cm height, the trapped dust weighed 105.39 kg; at 400 cm, however, it weighed only 26.75 kg. The sediment traps closest to the surface captured sand with a median diameter of 174 µm; at +15 cm this was 134 µm, at +200 cm 118 µm, and at +400 cm only 42 µm (coarse silt).

The total organic carbon (TOC_w_) and inorganic carbonates in the sediment traps closely reflected the influence of the height above the ground surface.

The success of the various diversifications varied over the years. However, disregarding 2019, when a drought prevented the normal development of cover crops, diversification D2 (with oat intercropped) mitigated the rate of wind erosion most efficiently—by 25%, i.e., from 2.6 t ha^−1^ y^−1^ to 1.9 t ha^−1^ y^−1^, compared to the asparagus monoculture.

Oat ensured a higher surface roughness and better soil coverage, and protected against wind erosion somewhat better than fodder pea. Dust generation and humus loss were also more limited under oat. However, this advantage is possibly compensated by the ability of pea as a leguminous plant to fix nitrogen. Ultimately, both cover crops seem to have a place in diversification planning.

## 4. Methods

The methodology of this research is described in detail in the Handbook compiled for the project [45]. In this paper, only those methods are presented which were judged relevant for the topic, i.e., soil moisture conservation and prevention of erosion by wind (see Section 4.1, Section 4.2, Section 4.3, Section 4.4, Section 4.5 and Section 4.6).

### 4.1. Estimation of Climatic Water Deficit

As shown by data from the closest principal meteorological station (Kecskemét, 46.91° N, 19.76° E, 112 m above sea level; registration number: 12970), situated at 25 km distance, the long-term (1880–2018) mean annual temperature is 10.7 °C and the annual average precipitation in the study area is 538 mm. The Thornthwaite annual potential evapotranspiration for 1951–2000 is ca. 750 mm (with unabated temperature rises, potential evaporation is increasing, and thus, water availability is deteriorating). In the years of the experiment, the annual temperature and precipitation amounted to 12.3 °C and 540.5 mm in 2018; 12.5 °C and 498.75 mm in 2019; and 11.9 °C and 587 mm in 2020. To establish evaporation levels, a Palmer-type one-layer bucket model used in the traditional calculation of the Palmer Drought Severity Index [46,47] was applied. The necessary soil parameters (water capacity in the field and at dewpoint, actual soil water reserves, root depth, etc.) were monitored in the Diverfarming project. The monthly potential evaporation (PET) was also computed from the data of nearby synoptic meteorological stations using the FAO and the Thornthwaite methods [48,49].

### 4.2. Crop Treatments

Given the limited choice of crops which can be grown in the region, an irrigated asparagus field of 1.3 ha area was selected for the experiment. Asparagus cultivation technology involves foil-covered, ca. 30 cm-high ridges [50] built mechanically parallel to the prevailing wind direction (Figure 7), thus enhancing wind channel effects and wind erosion.

Asparagus is a high-value, but also a highly labour- and cost-intensive crop. In recent years, both its harvest area and yields have shrunk considerably in Hungary; in 2006, (predominantly white) asparagus was cultivated in an almost 2000 ha area [50], while in 2021, this area was around 1400 ha. The yields ranged from 4.5 t ha^−1^ to 6.5 t ha^−1^; 60–90% were exported, mostly to Germany. A major reason for this decline is the lack of a labour force, intensified by the COVID-19 epidemic, and the low purchase prices paid to producers by traders. (Mechanized harvesting is technologically possible; the costs of machinery, however, far exceed the capacities of Hungarian farmers). 

In semiarid environments, irrigation is a crucial requirement for profitable asparagus cultivation. Campi, P., et al. [51] claimed that while full irrigation maximizes asparagus yields, reduced irrigation in critical phenological phases (i.e., 50% of evapotranspiration from the first fern flush until the end of the growing season), also provides acceptable yields. Therefore, in the study area, regulated deficit irrigation (RDI) was originally planned when asparagus was planted in 2010. The irrigation system operated for six years, during which, the perennial crop was well established; however, it was blocked just before the start of the experiment and, therefore, no further irrigation could be employed and rainfed cultivation was instead practiced.

Additional inputs also make asparagus production cost-effective. In the experiment, management activities began with the preparation of ridges in February or March, depending on weather (frost) conditions. The fern was cut and the ridges were removed in mid-June.

The nutrient demands of asparagus plantations in Hungary are summarized in the literature [52] as follows. To induce shoot (spear) growth, the uppermost 60 cm soil layer should include 250 mg P_2_O_5_ kg^−1^ and 300 mg K_2_O kg^−1^. For fertilization, this means the following amounts: nitrogen (NO^3−^, NH^4+^): 160 kg ha^−1^; phosphorus (P_2_O_5_): 46 kg ha^−1^; and potassium (K_2_O): 146 kg ha^−1^. Microelements (Ca, Mg, S, Cu, Zn, Mn) are optimally added as foliar fertilization.

Accordingly, in the study plots, NPK fertilization (around 300 kg ha^−1^) took place on two occasions per year (two-thirds of the annual dose in June, and one-third in August); additionally, calcium-, magnesium-, and sulphur-oxides were also applied.

Further treatments are necessary for weed control, beginning with the emergence of shoots in early June. Avoiding tillage and increasing soil health, quick-growing cover crops such as fescues, perennial ryegrass, and clover are planted in the alleys. At Jakabszállás, intercropping was established in April every year.

The need for plant protection is motivated by a great variety of pests affecting asparagus [53]: crown and root rot (***Fusarium* spp.**), fungal asparagus rust (***Puccinia* spp.**), grey mould (***Botrytis cinerea***), leaf spots (***Cercospora asparagi***), asparagus beetles (***Crioceris asparagi***), and others. At Jakabszállás, plant protection started with weeding by rolling or rotating in early June (as required), and then also involved the application of pesticides (with the effective substance metribuzine) in June, with a combination of fungicides (difenoconazole, azoxystrobin, dyoctil-potassium-sulfosuccinate, metiram, tebuconazole, and boscalid) with insecticide (alpha-cypermethrin) at the end of July. Fungicide and insecticide treatments were repeated in late August or early September, depending on the weather. In some years, additional mechanical weeding (by rolling or rotating) also became necessary. Occasionally, when beetles attacked, spraying with insecticide-generating emulsion (esfenvalerate and acetamiprid) could be undertaken. This brief summary illustrates the wide variety of necessary plant protection interventions in the asparagus field.

### 4.3. Crop Diversification Options

When selecting herbs for intercropping, it is important to avoid competition for water. In the experiment, two diversifications were introduced to raise the provision level of the ecosystem services: asparagus (*Asparagus officinalis* L.) monocrop (M, control) and plots diversified with fodder pea (*Pisum sativum* L.) (D1) and oat (*Avena sativa* L.) (D2) were sown in the alleys between ridges. Both the leguminous pea, which binds nitrogen from the atmosphere and is rich in phosphorus and potassium, and oat, which is also rich in nutrients and tolerant to weather extremes, are widely grown as fodder crops in the Kiskunság region. A comparison of root depths shows some overlaps, but the typical depth of the hair root zone is sufficiently different (Figure 8) and precludes strong competition [54].

### 4.4. Monitoring Soil Properties and Soil Moisture

In the Diverfarming project, the following soil properties were monitored [45]: bulk density (g cm^−3^); soil water content at a wilting point of 1500 kPa and at a field capacity of −33 kPa (cm^−3^ cm^−3^); soil texture (sand, silt and clay contents, %); pH; electrical conductivity (dS m^−1^); total organic carbon (mg g^−1^); nitrogen content (mg g^−1^); cation exchange capacity (cmol kg^−1^); sum of bases (cmol kg^−1^); carbonate content (%); rock fragments and gravels (>2 mm; %); and actual field moisture content (cm^−3^ cm^−3^). Monitoring techniques rely—among other sources—on Huber, S., et al. [55] and are described in the Handbook [45]. The experimental design involved strip-plots in each block with three replicates. The plot size was 160 × 8.4 = 1344 m^2^, including three asparagus rows with inter-rows. Crop (asparagus) samples from the central row were collected for each plot at harvest (every June over three crop cycles, 2018–2020) and again in autumn (October). In parallel, soil samples were taken from two layers (0–10 and 10–30 cm), with three composite samples per plot (nine samples per treatment).

The soil moisture content impacts many fundamental biophysical processes [56]: germination, plant growth, the microbial decomposition of the soil organic matter, and nutrient transformations in the root zone. Heat and water transfer at the land–atmosphere interface is also dependent on moisture content. For soil moisture monitoring, Time Domain Reflectrometry (TDR) was used (Decagon Devices Inc., Pullman, WA, USA). The sensors were placed at −10, −30, and −100 cm depths in the alleys between the ridges. This method is based on permittivity, which is the measure of a material’s ability to resist an electric field [57]. The soil moisture was monitored continuously over the three years of the experiment.

### 4.5. Wind Erosion Measurements

The amount of wind-transported sediment was measured in a 100 m cross-section in traps placed on the ground surface and at 30 cm, 200 cm, and 400 cm heights above the surface. The wind velocity data from the observations of the meteorological station were evaluated. The critical velocity for aeolian sand transport was established from the results of wind tunnel experiments [58,59]. Under the conditions of the Kiskunság sand region, previous research estimated a critical entrainment velocity of 6 ms^−1^ for a 10 cm height above the ground surface [25]. Although the critical velocity also varies with the soil organic matter content, in the case of blown-sand soils, it has no significance. In the deposited material, the total carbon and inorganic carbonate content were determined using the wet combustion technique [60].

### 4.6. Emission of Greenhouse Gases

In the Pannonian pedoclimatic region, half of the greenhouse gas (CO_2_, CH_4_ and N_2_O) emissions are derived from agriculture. Above a 10 °C soil temperature, CO_2_ emissions are continuous and N_2_O is released from nitrification and denitrification processes. On the other hand, soil is a CH_4_ sink, and methane is converted first into methanol (CH_3_OH) and finally to CO_2_ by methanotrophic bacteria.

For gas measurements performed on 25–33 occasions a year at a higher than +5 °C soil temperature, the static chamber method was applied, with timing adjusted to fertilization dates [61,62]. Parallel sampling was carried out at 0, 10, 20, and 30 min from a total of 18 closed chambers (Figure 9). CO_2_ and CH_4_ concentrations were measured by gas chromatography with a flame ionization detector (FID) and N_2_O with an electron capture detector (ECD). Fluxes in the soil were computed from the accumulation of concentrations and interpreted as a function of the weather and plant phenology.

## 5. Conclusions

The introduction of crop diversification in the Kiskunság sand region can involve the following environmental benefits: Higher organic matter content and cation exchange capacity point to increased soil fertility.More complete soil cover and reduced cultivation diminish desiccation hazards and ensure the necessary moisture in the topsoil for cover crops too.

Although measurements were limited to only three growing seasons, certain conclusions can perhaps still be drawn from them regarding soil fertility, soil moisture conservation, and wind erosion risk.
The condition of the soil surface was the same in all plots; the differences in sediment transport by the wind were exclusively caused by diversification.The particle size of the entrained dust fraction stabilized at a height of 400 cm, where the proportion of the 0.7–8 μm fraction (fine organic matter) increased. During periods of drought, dust storms significantly reduced soil fertility.Complete desiccation of the soil surface often occurred during the experimental period. In such cases, there was no solution to protect the soil, so diversification did not help either. At times of minimal wetting, crop diversity and the use of cover crops in broad alleys can reduce wind erosion.

## Figures and Tables

**Figure 1 plants-13-00494-f001:**
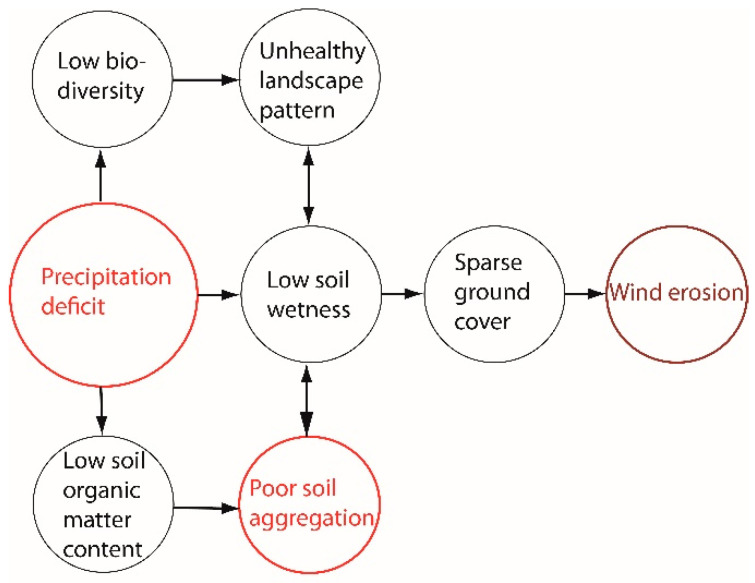
Environmental problems in blown-sand areas.

**Figure 2 plants-13-00494-f002:**
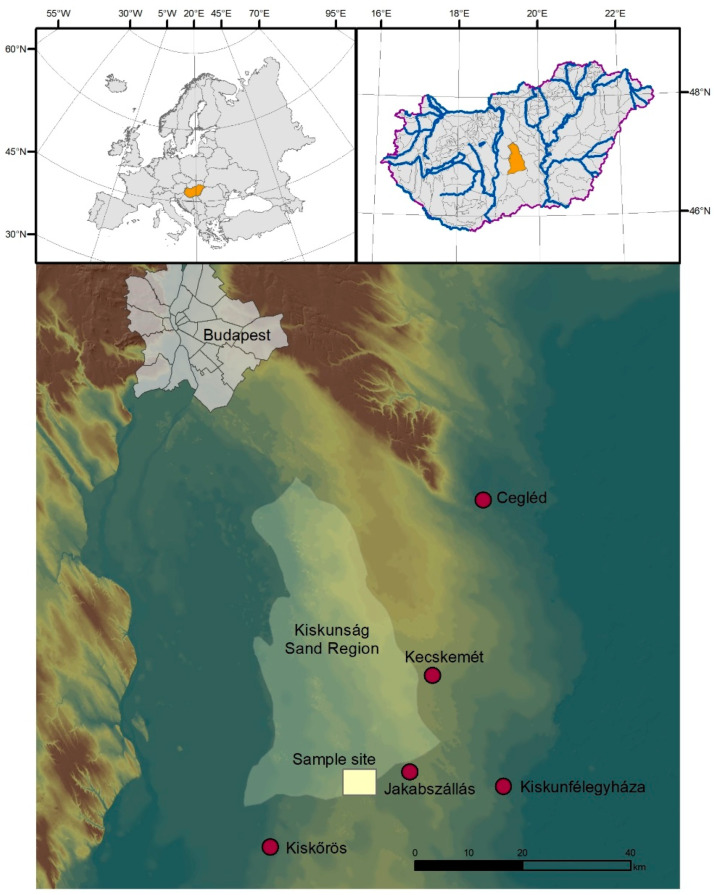
Location of the Jakabszállás study area in the Kiskunság sand region.

**Figure 3 plants-13-00494-f003:**
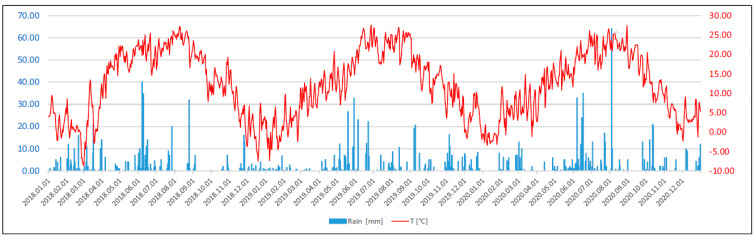
Daily mean air temperature (curve, °C) and rainfall data (columns, mm) over the three cycles of the experiment (2018–2020).

**Figure 4 plants-13-00494-f004:**
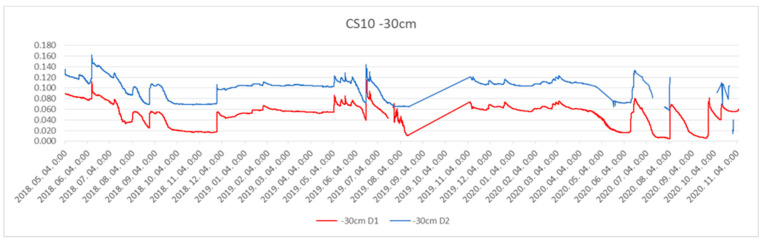
Actual soil moisture content (cm^3^ cm^−3^) at 10–30 cm depth. The upper curve is for D2 (asparagus + oat) and the lower is for D1 (asparagus + pea).

**Figure 5 plants-13-00494-f005:**
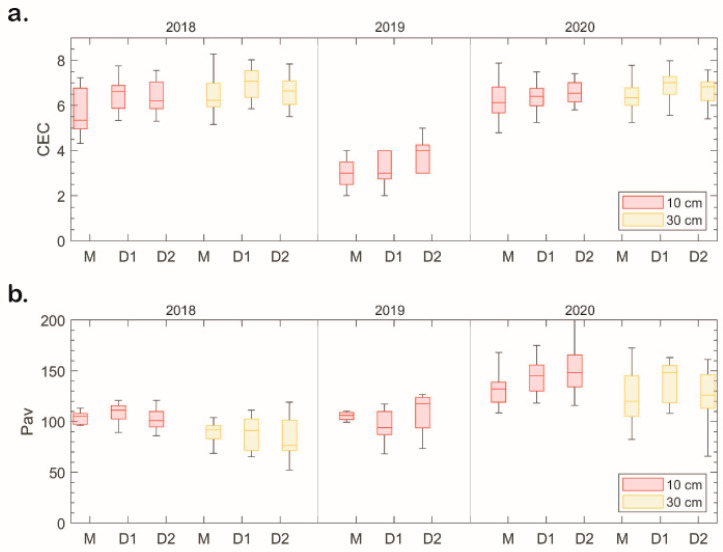
Cation exchange capacity (CEC; cmol kg^−1^) (**a**) and bioavailable phosphorus (P_av_; mg kg^−1^) (**b**) shown in a box diagram (upper and lower quartiles, median, and standard deviation) in the soil of the studied plots over the three-year period for the individual diversifications (M = asparagus monoculture; D1 = asparagus + field pea; D2 = asparagus + oat) in 0–10 cm and 10–30 cm soil layers.

**Figure 6 plants-13-00494-f006:**
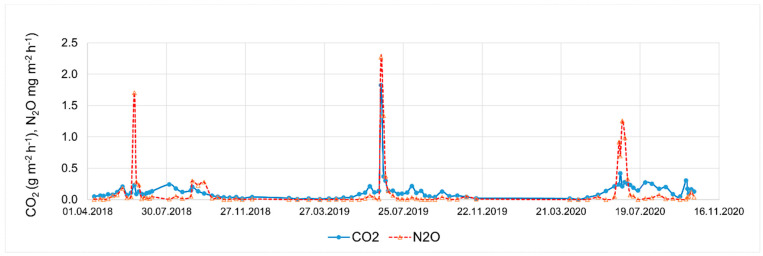
Soil greenhouse gases (CO_2_ and N_2_O) emissions (mg m^−2^ h^−1^) over the three years (by L. Horváth).

**Figure 7 plants-13-00494-f007:**
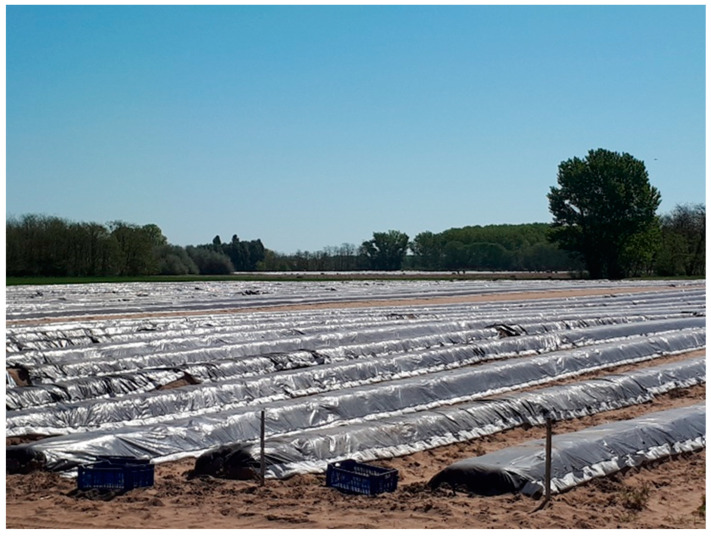
Asparagus ridges with black plastic foil cover in the Jakabszállás study area, 21 April 2018 (by J. Dezső).

**Figure 8 plants-13-00494-f008:**
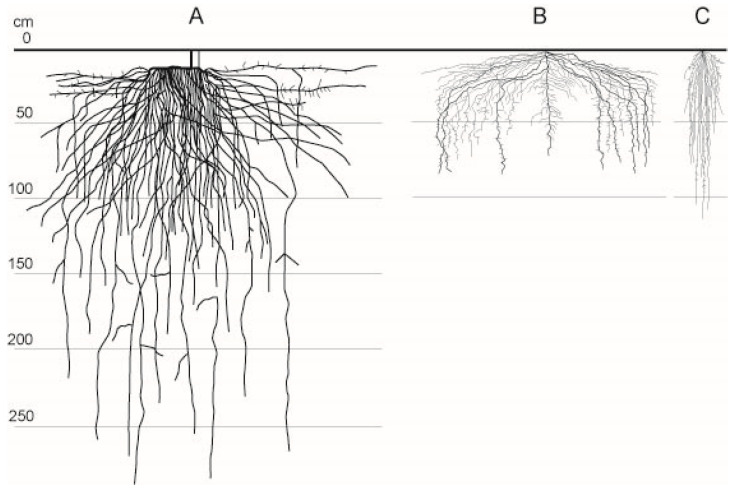
Typical root system of asparagus (**A**), field pea (**B**), and oat (**C**) (D. Lóczy after Soil and Health Library, Chudleigh, Tasmania, Australia, https://soilandhealth.org/agricultural-library/ accessed on 11 February 2023).

**Figure 9 plants-13-00494-f009:**
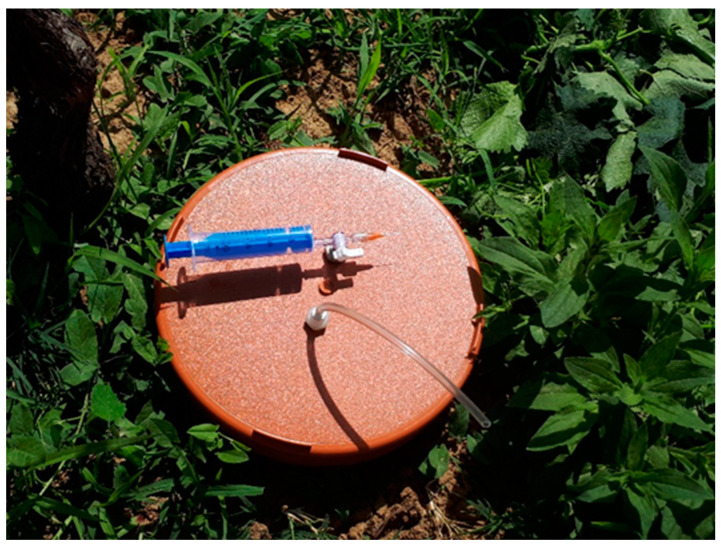
Closed-system static chamber for gas sampling with injection syringe and needle (photo by L. Horváth).

## Data Availability

Papers related to the Diverfarming project are uploaded to the Zenodo repository: https://zenodo.org/communities/diverfarming?q=&l=list&p=1&s=10&sort=newest (accessed on 6 July 2023). However, datasets are embargoed and only available to project participants for three years.

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
