# Peer review of "Soil Moisture Conservation through Crop Diversification and Related Ecosystem Services in a Blown-Sand Area with High Drought Hazard"

_plants, 2024, doi:10.3390/plants13040494_

Round 1

Reviewer 1 Report

Comments and Suggestions for Authors

Dear Editor of Plants,

I have reviewed the manuscript Soil moisture conservation through crop diversification and related ecosystem services in a blown-sand area with high drought hazard”.

The authors evaluated the impacts of intercropping asparagus with different herbs on some provisioning and regulating ecosystem services were evaluated in the Kiskunság sand regions, in the Pannonian region.

The findings highlight important information about the behavior of greenhouse gases and sediment yield by wind erosion. Those results evidence that strategies for the diversification of crops are necessary and efficient to maintain and increase the cover of the soil and, as a consequence, avoid soil erosion by wind, mainly in regions with fragile soils. The use of native plants would be an interesting strategy to test since these plants are adapted to edaphoclimatic conditions.

Thus, the paper has a relevant subject, adequate to be published in the journal Plants.

The whole manuscript is well-structured. The Figures are well-designed.

All sections are well-described, clear, and easy to understand. However, several important information are missing.

What is the hypothesis and objective of the study? The proposed actions do not indicate that this will be done, since the authors are already indicating what does not work or cannot be done.

Although the article has important information for sandy and fragile soils, it is necessary to have a clear objective, to describe the entire methodology of all the results that were presented.

The results and discussions must be described according to the information that appears in the methodology.

Conclusions must be made according to the objectives and results presented. It is not possible to conclude about something that has not been evaluated. Some conclusions were described, but the study that generated the conclusions does not appear in the objectives, methodology, or results.

Minor comments

Regulated deficit irrigation was one strategy used in the experiment and should be better explored in the Introduction section.

Lines 236-237: Insert the scientific name of all species.

Line 248: Which is the water tension applied at field capacity?

Line 306: When the boundaries of the soil layer are indicated, the correct term is soil layer, instead of soil depth.

Line 322: Indicates on the figure if there are differences among treatments.

Line 331: Since decomposition is low, the residues can provide soil cover and protection to avoid soil erosion.

Line 373: Have mycorrhizae been evaluated? If not, this information cannot be in the conclusion.

Line 377: Were the socioeconomic conditions assessed?

Comments on the Quality of English Language

The quality of the English Language is good.

Reviewer 2 Report

Comments and Suggestions for Authors

Congratulations on your well-organized, easy-to-understand and readable work. An innovative approach to work and use of corn influencing the optimization of work when sowing and sowing corn. Soil moisture reserves are a key factor in maintaining soil fertility and all other related ecosystem services. In the semi-arid, sand-blown areas subjected to drainage Water conservation is a particularly important task for agriculture. A three-year experiment was conducted in the Pannonian region to assess the impact of growing asparagus with various herbs on some of the ecosystem services that provide supply and regulation in the sandy regions of Kiskunság. Advice for farmers was formulated based on findings based on a number of measured soil physical and chemical parameters, as well as on yield and quality characteristics. The topic is very interesting, the purpose and implementation of the article are clear, and the information provided is informative. The contribution of the article is obvious and it will certainly become the work of researchers in this field. This work contains some technical issues that need to be resolved before the manuscript is ready for publication.

Comments on the Quality of English Language

Congratulations on your well-organized, easy-to-understand and readable work. An innovative approach to work and use of corn influencing the optimization of work when sowing and sowing corn. Soil moisture reserves are a key factor in maintaining soil fertility and all other related ecosystem services. In the semi-arid, sand-blown areas subjected to drainage Water conservation is a particularly important task for agriculture. A three-year experiment was conducted in the Pannonian region to assess the impact of growing asparagus with various herbs on some of the ecosystem services that provide supply and regulation in the sandy regions of Kiskunság. Advice for farmers was formulated based on findings based on a number of measured soil physical and chemical parameters, as well as on yield and quality characteristics. The topic is very interesting, the purpose and implementation of the article are clear, and the information provided is informative. The contribution of the article is obvious and it will certainly become the work of researchers in this field. This work contains some technical issues that need to be resolved before the manuscript is ready for publication.

Reviewer 3 Report

Comments and Suggestions for Authors

This manuscript studied the effects of crop diversification on soil properties and greenhouse gas emissions in a blown-sand area. The topic is interesting. However, there were lots of introduction, but less results to supports the conclusions. The major comments are as follows.

1.     In the introduction, the specific hypotheses or objectives of this study should be given.

2.     In the methods part, more details of the experimental plots, experimental period, and sampling frequency should be given.

3.     In the results parts, there were only soil moisture, soil CEC, soil available P, soil CO2 and N2O emission data were shown. Besides, these data were only for one or two treatments. It is not enough to assess the effects of crop diversification.

4.     The conclusions were not supported by the results.

Round 2

Reviewer 1 Report

Comments and Suggestions for Authors

Dear Editor of Plants,

I have reviewed the revised manuscript Soil moisture conservation through crop diversification and related ecosystem services in a blown-sand area with high drought hazard”.

The authors evaluated the impacts of intercropping asparagus with different herbs on some provisioning and regulating ecosystem services were evaluated in the Kiskunság sand regions, in the Pannonian region.

The authors made all corrections and improvements in the manuscript, as recommended by the both reviewers.

The paper has a relevant subject, adequate to be published in the journal Plants. Thus, I recommend accepting the paper for publication at this present form.

Reviewer 3 Report

Comments and Suggestions for Authors

The authors have revised the manuscript according to the reviewers' comments. I recommend to accept it.